# Design and Experimental Investigation of a Transplanting Mechanism for Super Rice Pot Seedlings

**Maile Zhou \*, Zhaoxiang Wei, Zeliang Wang, Hao Sun, Guibin Wang and Jianjun Yin**

School of Agricultural Engineering, Jiangsu University, Zhenjiang 212013, China;
2222216053@stmail.ujs.edu.cn (Z.W.); 2222116032@stmail.ujs.edu.cn (Z.W.); 2222216061@stmail.ujs.edu.cn (H.S.);
2212216052@stmail.ujs.edu.cn (G.W.); yinjianjun@ujs.edu.cn (J.Y.)
* Correspondence: zhoumaile@ujs.edu.cn

**Abstract:** Super rice contains a variety of advantageous characteristics. However, current rice seedling transplanting machines fail to achieve the necessary trajectory and distance required for super rice mechanized transplanting. To address this issue, this study introduces a differential-speed rotary mechanism for transplanting super rice pot seedlings. The developed mechanism operates using a non-uniform speed differential gear train, which enables the transplanting arm components to mimic the specific trajectory and posture necessary for transplanting super rice pot seedlings. The kinematic model of the differential-speed rotary super rice pot seedling transplanting mechanism (PSTM) was established, and optimization design software was developed. This software facilitated the determination of a set of mechanism parameters optimized for super rice pot seedling transplantation. The results obtained from virtual simulations were found to be in alignment with those from the optimization software, thereby verifying the accuracy of the theoretical analysis and simulation. A testing bench for the rice PSTM was also developed and used for pot seedling pickup experiments. The bench tests demonstrated that the designed super rice PSTM yielded a seedling pickup success rate of 97% and a seedling injury rate of 1.8% when operating at an efficiency of 200 times/min.

**Keywords:** transplanting mechanism; super rice; pot seedlings; non-circular gear; trajectory and posture

## 1. Introduction

Rice, being a critical food crop, plays a pivotal role in ensuring global food security. Therefore, enhancing its production is of utmost importance. In the 21st century, research in the field of super rice in China has shown notable growth, with China leading the world in research on super high-yielding rice breeding [1]. There are primarily three methods employed in rice cultivation: transplanting, pot seedling transplanting, and direct seeding. Direct seeding technology, which involves sowing rice seeds directly into the field, saves significant time and labor costs. However, this method necessitates extensive management and requires a high degree of professional expertise from the planters. Moreover, it demands specific regional and climatic conditions and may not be well suited for small field agronomy [2,3]. Conventional rice transplanting uses blanket seedlings, where a certain number of seedlings are transplanted into the paddy field by tearing, causing significant damage to the seedlings. The extended seedling survival stage weakens the advantage of increased production and hampers rice growth [4,5]. On the other hand, rice pot seedling transplanting involves transplanting seedlings with independent seedling pot substrates into the rice field. The seedlings have their own nutritional pots, thus ensuring that the roots are not damaged during transplantation. Furthermore, the absence of a seedling survival stage allows for earlier tillering, extending the effective growth period of the crop. This method has been shown to foster early and rapid growth in rice, increase effective tillering, and ultimately boost rice yield [6–8].

In the 1990s, Japan pioneered the production of a top-out rice pot seedling transplanter. This transplanter sequentially performed operations such as pot seedling extraction, trans-

portation, and planting. However, this solution was marred by its complex and inefficient mechanical structure. A five-bar seedling pickup mechanism, subsequently developed by South Korea, was made up of connecting rod sliders. Despite its simplicity, this structure was limited by the substantial inertial force of the connecting rod mechanism, leading to large vibrations during high-speed operation. Consequently, it was only suitable for low-speed transplanting [9]. Ye et al. [10] proposed a transplanting mechanism incorporating incomplete eccentric gear trains. Their theoretical analysis, virtual simulations, and high-speed camera kinematics test results affirmed the validity of their model and design. The optimized transplanting mechanism showed potential for application in rice pot seedling transplanting machines. Wu et al. [11] suggested a method for determining the pitch curve of non-circular gears. This approach altered the local absolute motion trajectory and then computed the relative motion trajectory. A three-arm rotary rice pot seedling transplanter mechanism (PSTM) was designed, meeting the stringent vertical transplanting needs of rice pot seedlings and validating the design of the three-arm rotary rice PSTM. Yu et al. [12] introduced a rotary rice PSTM, using an elliptical incomplete non-circular gear as the transmission mechanism. They optimized a set of structural parameters conducive for rice pot seedling transplantation, offering a reference point for future rotary rice PSTM designs. Cai et al. [13], acknowledging the limitation of existing rice pot seedling transplanters, which could not achieve variable row spacing transplantation, proposed a device using a movable seedling transfer tube. This device could alter the distance between transplanting rows without affecting the seedling picking distance of the seedling carrying platform. By analyzing the seedling pickup trajectory and pot seedling pickup characteristics, a link-type seedling picking mechanism was proposed. This mechanism consisted of a double crank mechanism and a crank rocker mechanism. The mechanism demonstrated an average seedling pickup success rate of 89.96% and an average seedling fall rate of 3.45%. Sun et al. [6] targeted the constraints of trajectory shape and attitude design in the transplanting mechanism of a single planetary carrier and two-stage gear transmission when transplanting rice pot seedlings. They proposed a degree-of-freedom label diagram screening criterion, capable of solving the planetary gear train mechanism for a complex trajectory. This approach overcomes existing configuration limitations, offering a novel and feasible design for the rice PSTM. In their quest to ensure the planetary gear train transplanting mechanism met the complex spatial trajectory and fulfilled the agronomic requirements of transplanting rice seedlings in wide and narrow row pots, Wang et al. [14] proposed a reverse design method for a transplanting mechanism of inconstant velocity planetary gear trains based on a general space continuous closed trajectory. They established a kinematics model of the space open-chain 2R mechanism and acquired the parameters of the transplanting mechanism by problem solving.

The existing conventional rice pot seedling transplanters are mainly designed for conventional rice transplants with seedling heights of around 140–160 mm and plant spacing of around 140–180 mm. For super rice, when the seedling height is greater than 170 mm, the required plant spacing is 180–240 mm. Conventional rice pot seedling transplanting cannot meet the requirements of mechanized transplanting of super rice. Therefore, a transplanting mechanism that can meet the requirements of transplanting trajectory and spacing of super rice pot seedlings has been developed. An examination of both domestic and international research suggests that non-circular gear transmission could offer the benefits of variable speed transmission ratio, stable operation, and minimal vibration [15,16]. In this study, we aimed to address the transplanting of super rice pot seedlings by developing a double-arm differential-speed non-circular gear system for the super rice PSTM. This system employs a single-side transplanting arm with a three-stage gear transmission. During the operation, two transplanting arms alternately perform seedling pickup, conveyance, and planting, completing two transplanting actions within a single working cycle.

## 2. Materials and Methods

### 2.1. Working Principle

The super rice PSTM comprises a differential non-circular gear train and two transplanting arms, as depicted in Figure 1. This differential non-circular gear train includes seven non-circular gears and a planetary carrier, which collectively have two degrees of freedom. The sun gear and the planetary carrier serve as the primary moving parts and rotate in the same direction but at differential speeds. The sun gear meshes on both sides with intermediate gear I and intermediate gear III simultaneously, while the two sides of intermediate gear II engage with planetary gear I and intermediate gear I, rotating individually. Similarly, both sides of intermediate gear IV mesh with planetary gear II and intermediate gear III, again rotating individually. The sun gear drives planetary gears I and II to rotate via three-stage gear transmission, subsequently driving the rotation of transplanting arms I and II, which are fixed to the planet shafts I and II, respectively. As the planetary carrier rotates around its center of rotation, it drives intermediate gears I, II, III, and IV and planetary gears I and II, which are hinged to it, to execute an epicyclic motion along the center of rotation of the planetary carrier. This rotation propels the transplanting arms I and II to perform a revolving movement. The combination of these two movements enables the transplanting arms to generate the trajectory and posture required for transplanting rice pot seedlings.

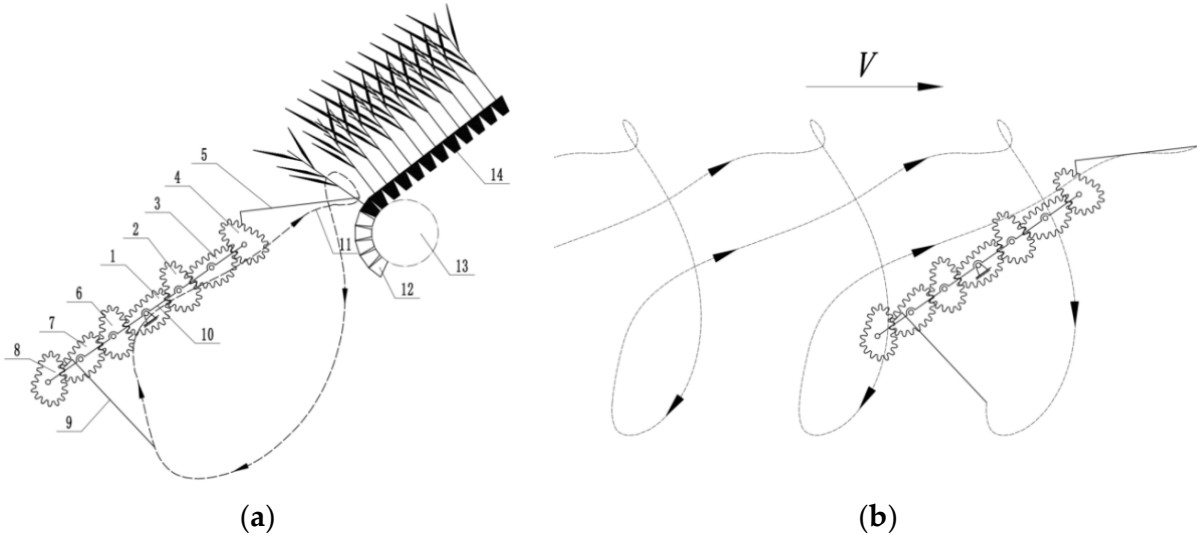

(**a**)                                          (**b**)

**Figure 1.** Working principle diagram of super rice pot seedling transplanter mechanism (PSTM). 1. Sun gear; 2. Intermediate gear I; 3. Intermediate gear II; 4. Planetary gear I; 5. Transplanting arm I; 6. Intermediate gear III; 7. Intermediate gear IV; 8. Planetary gear II; 9. Transplanting arm II; 10. planetary carrier; 11. relative motion trajectory; 12. seedling tray; 13. conveying roller; 14. pot seedlings. (**a**) Relative Motion. (**b**) Absolute Motion.

### 2.2. Composition

The transplanting arm comprises several components: a shell, seedling clamping slices, a seedling pushing rod, a seedling clamping device, a seedling pushing device, a cam, and a shifting fork. The cam is affixed to the planetary carrier, while the shifting fork is hinged to the shell of the transplanting arm. The transplanting arm shell rotates relative to the cam (planetary carrier) with the planetary gear, and the cam and shifting fork together form a swing follower cam mechanism. The end of the shifting fork, located away from the cam, governs the pusher rod, facilitating a reciprocating linear motion. Both the seedling clamping device and the seedling pushing device are rigidly connected to the pusher rod. A seedling clamping device is arranged on the outer side of a pair of seedling clamping slices, and the seedling pushing device is positioned under these slices. In the initial state, the seedling clamping device does not constrain the seedling clamping

slices, allowing them to open naturally due to their inherent elasticity. Upon reaching the seedling clamping position, the seedling pushing rod propels the seedling clamping device to retract, causing an upward movement that quickly compresses the clamping slices from both sides, thus securing the seedlings within. When the mechanism reaches the pushing position, the shifting fork proceeds to the cam's return stage, swiftly projecting the seedling pushing rod under the influence of the pushing spring. The forward movement of the seedling clamping device, rigidly connected with the seedling pushing rod, ceases to exert compressive force on the seedling clamping slices. The clamping slices rapidly open under their own elasticity, while simultaneously, the seedling pushing device, affixed at the front end of the seedling pushing rod, interacts with the seedling pot matrix to thrust the seedlings into the paddy field.

*2.3. Kinematic Analysis*

Bezier curves have many advantages, such as high accuracy, good controllability, high smoothness, and simple calculation. Research has shown that designing non-circular gears through Bessel curves has the advantages of multiple parameters and a wide range of unequal speed transmission, which meets the requirements of super rice transplanting mechanisms for unequal speed transmission. This study utilizes a three-stage differential-speed non-circular gear epicyclic gear train as the object of research, where the non-circular gear adopts a Bezier gear and its gear pitch curve forms a Bezier curve [17]. A theoretical kinematic model for the super rice pot seedling transplanting mechanism was established. With the sun gear's rotation center as the coordinate origin, we applied the Bezier curve fitting equation to derive the polar coordinate equation of the sun gear pitch curve. Considering that the average transmission ratio of two meshing non-circular gears equals one, the center distance is determined. The polar coordinate equation of the pitch curve for the conjugate gear is calculated based on the center distance and the instantaneous transmission ratio, and this principle is utilized to compute the pitch curve equation of each gear. Eventually, a kinematic model of the cusp point of the transplanting arm, which varies with the parameters of the sun gear pitch curve, is established [18]. Figure 2 illustrates the schematic diagram of the initial position of the transplanting mechanism.

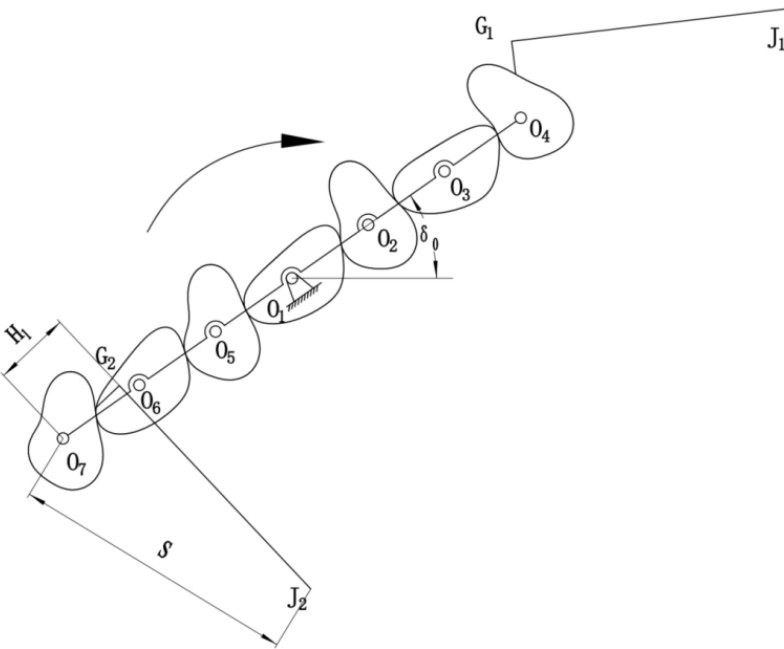

**Figure 2.** Schematic diagram of the initial position of the transplanting mechanism for super rice pot seedlings.

In this study, the polar coordinate equation of the sun gear pitch curve is given by

$$
\begin{cases}
r_1(i) = \sqrt{[p_x(i)]^2 + [p_y(i)]^2} \\
\theta_1(i) = arctan \frac{p_y(i)}{p_x(i)}
\end{cases}
\tag{1}
$$

where: $r_1(i)$—Polar coordinate radius of the sun gear pitch curve;

$\theta_1(i)$—Polar coordinate angle of the sun gear pitch curve.

Utilizing the dichotomy method (Ye et al., 2015), the center distance of non-circular gear transmission is obtained from the sun gear pitch curve data:

$$
\eta = \int_0^{2\pi} \frac{r_1(i)}{a_3 - r_1(i)} d_{\theta(i)} = 2\pi
\tag{2}
$$

where: $\eta$—Rotation angle of the driven gear after one rotation of the driving gear;

$a_3$—Center distance calculation intermediary parameters.

The polar coordinate equation of the Intermediate Gear 1 Curve is

$$
\begin{cases}
r_2(i) = l_{o_1 o_2} - r_1(i) \\
\theta_2(i) = \int_0^{\theta_1(i)} \frac{r_1(j)}{r_2(j)} d_{\theta_1}(j)
\end{cases}
\tag{3}
$$

where: $r_{2(i)}$—Polar coordinate radius of the pitch curve of Intermediate Gear 1;

$\theta_{2(i)}$—Polar angle of the pitch curve of Intermediate Gear 1.

The polar coordinate equation of Intermediate Gear 2 Curve is

$$
\begin{cases}
r_3(i) = l_{o_2 o_3} - r_2(i) \\
\theta_3(i) = \int_0^{\theta_2(i)} \frac{r_2(j)}{r_3(j)} d_{\theta_2}(j)
\end{cases}
\tag{4}
$$

where: $r_4(i)$—Polar coordinate radius of the pitch curve of Planetary Gear 1;

$\theta_4(i)$—Pitch curve polar coordinate angle of Planetary Gear 1.

The absolute rotation angles for each gear are calculated as follows:

It is assumed that the planetary carrier rotates clockwise relative to the frame (taken as the negative direction) and that the sun gear rotates $2\phi_H(i)$ degrees clockwise relative to the frame.

The rotation angle of the sun gear relative to the planetary carrier is

$$
\phi_{1H}(i) = -\phi_H(i)
\tag{5}
$$

The rotation angle of Intermediate Gear 1 relative to the planetary carrier is

$$
\phi_{2H}(i) = \int_0^{\phi_{1H}(i)} \frac{r_1(j)}{r_2(j)} d_{\phi_{1H}(j)}
\tag{6}
$$

The rotation angle of Intermediate Gear 2 relative to the planetary carrier is

$$
\phi_{3H}(i) = -\int_0^{\phi_{2H}(i)} \frac{r_2(j)}{r_3(j)} d_{\phi_{2H}(j)}
\tag{7}
$$

The rotation angle of Planetary Gear 1 relative to the planetary carrier is

$$
\phi_{4H}(i) = \int_0^{\phi_{3H}(i)} \frac{r_3(j)}{r_4(j)} d_{\phi_{3H}(j)}
\tag{8}
$$

The absolute rotation angle of the planetary carrier is

$$
\phi_0(i) = \phi_{H_0} - \phi_H(i)
\tag{9}
$$

The absolute rotation angle of the sun gear is

$$\phi_1(i) = \phi_0(i) + \phi_{1H}(i) = \phi_{H_0} - 2\phi_H(i) \tag{10}$$

The absolute rotation angle of Intermediate Gear 1 is as follows:

$$\phi_2(i) = \phi_0(i) + \phi_{2H}(i) = \phi_{H_0} - \phi_H(i) + \int_0^{\phi_{1H}(i)} \frac{r_1(j)}{r_2(j)} d_{\phi_{1H}(j)} \tag{11}$$

The absolute rotation angle of Intermediate Gear 2 is

$$\phi_3(i) = \phi_0(i) + \phi_{3H}(i) = \phi_{H_0} - \phi_H(i) - \int_0^{\phi_{2H}(i)} \frac{r_2(j)}{r_3(j)} d_{\phi_{2H}(j)} \tag{12}$$

The absolute rotation angle of Planetary Gear 1 is

$$\phi_4(i) = \phi_0(i) + \phi_{4H}(i) = \phi_{H_0} - \phi_H(i) + \int_0^{\phi_{3H}(i)} \frac{r_3(j)}{r_4(j)} d_{\phi_{3H}(j)} \tag{13}$$

The centers of rotation for each gear are computed as follows:
The center of rotation of the sun gear is

$$\begin{cases} x_{01} = 0 \\ y_{01} = 0 \end{cases} \tag{14}$$

The center of rotation of Intermediate Gear 1 is

$$\begin{cases} x_{02}(i) = l_{o_1 o_2} \cdot cos[\phi_0(i)] \\ y_{02}(i) = l_{o_1 o_2} \cdot sin[\phi_0(i)] \end{cases} \tag{15}$$

The center of rotation of Intermediate Gear 2 is

$$\begin{cases} x_{03}(i) = x_{02} + l_{o_2 o_3} \cdot cos[\phi_0(i)] \\ y_{03}(i) = y_{02} + l_{o_2 o_3} \cdot sin[\phi_0(i)] \end{cases} \tag{16}$$

The center of rotation of Planetary Gear 1 is

$$\begin{cases} x_{04}(i) = x_{03} + l_{o_3 o_4} \cdot cos[\phi_0(i)] \\ y_{04}(i) = y_{03} + l_{o_3 o_4} \cdot sin[\phi_0(i)] \end{cases} \tag{17}$$

The relative trajectory of the transplanting arm is determined as follows:
The relative motion coordinates of the tip point (Point J) of the transplanting arm are

$$\begin{cases} x_J(i) = x_{04}(i) + s \cdot cos[\phi_0(i) + \phi_{4H}(i) + \delta_0] \\ y_J(i) = y_{04}(i) + s \cdot sin[\phi_0(i) + \phi_{4H}(i) + \delta_0] \end{cases} \tag{18}$$

The relative motion coordinates of the inflection point (Point G) of the transplanting arm are

$$\begin{cases} x_G(i) = x_{04}(i) + H_1 \cdot cos[\phi_0(i) + \phi_{4H}(i) + \delta_0] \\ y_G(i) = y_{04}(i) + H_1 \cdot sin[\phi_0(i) + \phi_{4H}(i) + \delta_0] \end{cases} \tag{19}$$

The absolute trajectory of the transplanting arm is calculated as follows:
The correlation between the forward distance and the relative rotation angle of the planetary carrier during the operation of the transplanter is

$$H_2 = H \cdot \frac{\phi_H(i)}{\pi} \tag{20}$$

The absolute motion coordinates of the tip point (Point J) of the transplanting arm are

$$
\begin{cases}
x'_J(i) = x_j(i) + H_2 = x_{04}(i) + s \cdot cos[\phi_0(i) + \phi_{4H}(i) + \delta_0] + H \cdot \frac{\phi_H(i)}{\pi} \\
y'_J(i) = y_j(i) = y_{04}(i) + s \cdot sin[\phi_0(i) + \phi_{4H}(i) + \delta_0]
\end{cases}
\tag{21}
$$

The absolute motion coordinates of the inflection point (Point G) of the transplanting arm are

$$
\begin{cases}
x'_G(i) = x_G(i) + H_2 = x_{04}(i) + H_1 \cdot cos[\phi_0(i) + \phi_{4H}(i) + \delta_0] + H \cdot \frac{\phi_H(i)}{\pi} \\
y'_G(i) = y_G(i) = y_{04}(i) + H_1 \cdot sin[\phi_0(i) + \phi_{4H}(i) + \delta_0]
\end{cases}
\tag{22}
$$

The trajectories and postures formed by Transplanting Arm I and Transplanting Arm II are identical, differing only in phase. The motion equation of Transplanting Arm II can be obtained by rotating Transplanting Arm I with the planetary carrier through 180 degrees.

### 2.4. Parameter Optimization

#### 2.4.1. Compilation of Optimization Software

Leveraging the developed kinematic theoretical model, an optimal design software for the transplanting mechanism of super rice pot seedlings was constructed. This software mainly includes a menu bar, graphic display area, input parameter area, and target area. The graphic display area can display the schematic diagram and motion trajectory of the differential-speed rotary super rice PSTM in real time. The input parameter area is the mechanism parameters of the rice PSTM. The target area displays the various goals of the rice PSTM in the form of a progress bar. Due to the ambiguity of the goals, designers can judge the quality of the goals based on the status of the progress bar.

The main parameters that this software can input include the parameters of noncircular gears, initial installation angle of planetary carrier, structural parameters of transplanting arm, etc. The input parameters are parameters that have an impact on the working trajectory and posture of the super rice transplanting mechanism. The determination of the parameters is based on the structural characteristics of the existing transplanting machine and the agronomic requirements of the super rice transplanting. The software comprises a total of 31 parameters and 10 targets. The transplanting mechanism constitutes a differential-speed epicyclic gear train made up of seven non-circular gears (one sun gear, four intermediate gears, and two planetary gears). The installation positions of the two transplanting arms in this mechanism exhibit a phase angle difference of 180°, thereby enabling the transplanting of seedlings in two pots to be completed within a single rotation. The interface of the optimized software is depicted in Figure 3.

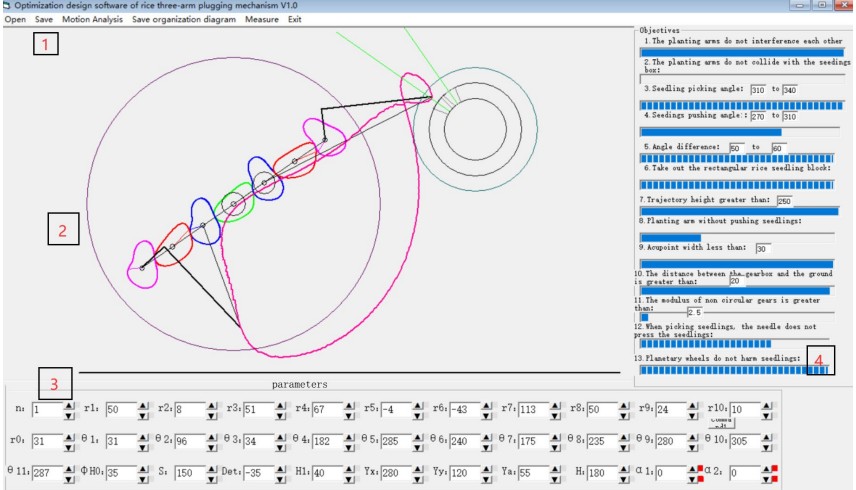

**Figure 3.** Optimal design software of super rice PSTM. 1. Menu bar; 2. graphic display area; 3. input parameter area; 4. target area.

### 2.4.2. Determination of Optimization Goals

In conjunction with the actual conditions of transplanting operations, the agronomic and mechanical design requirements were transformed into numerical optimization goals. These provide theoretical constraints for the automatic selection of parameters in the optimization design software [19] as follows:

(1) Non-interference of the transplanting arm is ensured, as illustrated in Figure 4.

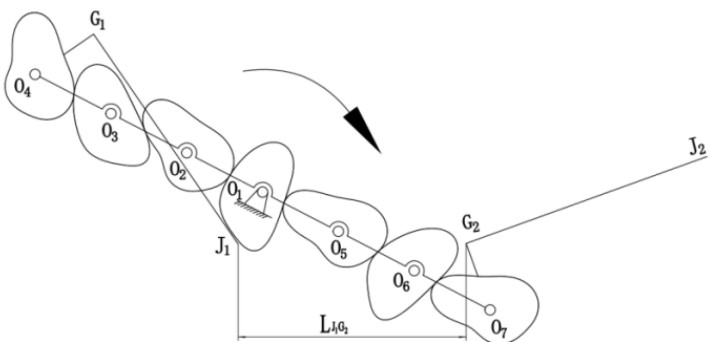

**Figure 4.** Schematic diagram of the non-interference of the transplanting arm.

Set $l_{J_1 G_2} \geq 5$ mm as an optimized condition for non-interference of the transplanting arm.

$$l_{J_1 G_2} = \sqrt{\left(x_{J_1} - x_{D_1}\right)^2 + \left(y_{J_1} - y_{D_1}\right)^2} \tag{23}$$

(2) The gear modulus must exceed 2.5. This condition ensures the bending fatigue strength of the non-circular gear's dedendum, implying that the minimal modulus of the non-circular gear shall exceed 2.5, that is $m_{min} \geq 2.5$.

(3) The seedling extraction angle ranges between $-5°$ and $10°$. As shown in Figure 5, this stage involves seedling extraction by the transplanting arm. The angle between the line on which the transplanting arm $G_1 J_1$ is located and the horizontal line passing through the inflection point $G_1$ of the transplanting arm is defined as seedling picking angle $\xi_1$. During the seedling clamping stage, it is required that the transplanting arm clamp should be as horizontal or tilted up as possible close to the seedlings. To minimize damage caused by contact between the transplanting arm and the seedling blade, it is crucial to ensure that the seedling extraction angle lies between $-5°$ and $10°$. Therefore, set $-5° \leq \xi_1 \leq 10°$.

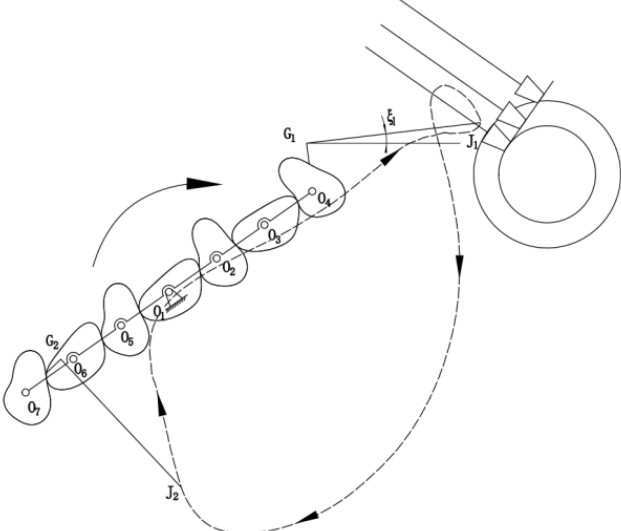

**Figure 5.** Schematic diagram of the angle of seedling picking of the transplanting arm.

(4) The pulled-out seedlings' height exceeds 25 mm. As depicted in Figure 6, to ensure that the transplanting arm can entirely lift the soil pot beneath the seedlings from the seedling tray, the pulled-out seedlings' height should be no less than 25 mm.

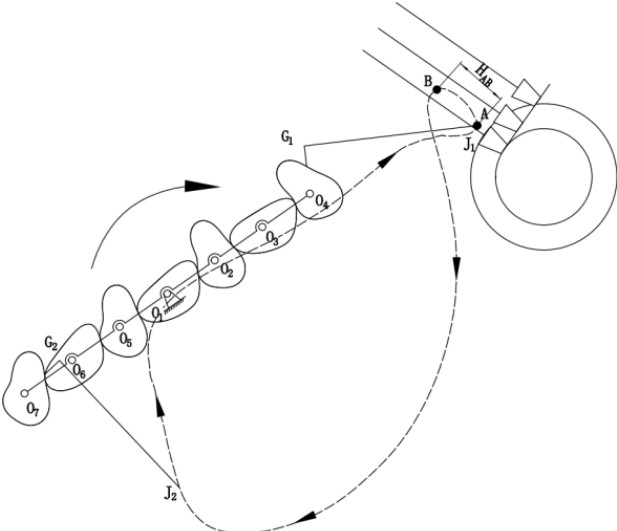

**Figure 6.** Schematic diagram of the height of the pulled-out seedlings.

(5) During seedling transportation, the seedlings should not interfere with the conveying rollers. As seen in Figure 7, the center of the conveyor roller is designated as point P, with its abscissa at the center of the circle being $x_p$ and the vertical coordinate of the circle center being $y_p$. The distance from point $J_1$ of the transplanting arm to the center of the conveying roller is recorded as $l_{J_1P}$. To circumvent damage instigated by the contact between the seedlings held by the transplanting arm and the conveying roller during the transport process, set

$$l_{J_1P} = \sqrt{\left(x_{J_1} - x_p\right)^2 + \left(y_{J_1} - y_p\right)^2} \geq r_p + k_1 \tag{24}$$

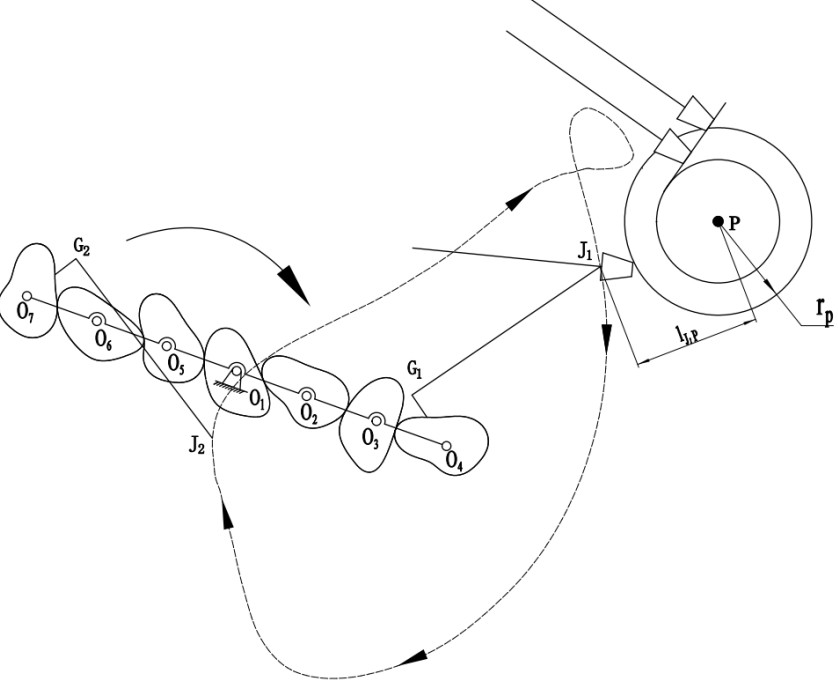

**Figure 7.** Schematic diagram of the stage of transporting seedlings.

In the equation, $r_p$ signifies the radius of the conveying roller. The radius of the conveying roller that pairs with the transplanting machine is 60 mm.

$k_1$ represents the height of the seedling's pot body, which is generally assumed to be 20 mm.

(6) The seedling pushing angle lies between 45° and 70°, and the angle difference ranges from 50° to 60°. As illustrated in Figure 8, the angle between the line where the transplanting arm $G_1J_1$ is located and the horizontal line passing through the point $J_1$ of the transplanting arm defines the seedling pushing angle $\xi_2$. To ensure upright planting, the pushing angle is $45° \leq \xi_2 \leq 70°$, and the angle difference is $50° \leq \xi_2 - \xi_1 \leq 60°$.

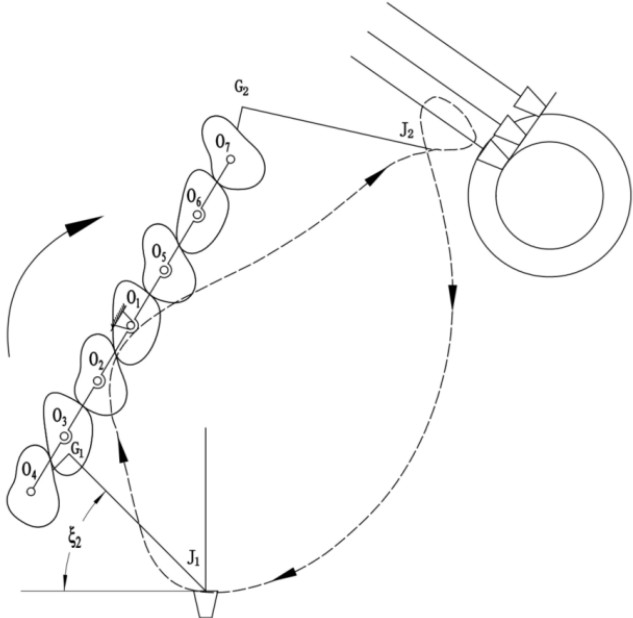

**Figure 8.** Schematic diagram of seedling pushing process.

(7) The trajectory height exceeds 300 mm. As shown in Figure 9, we define the height of the transplanting trajectory as $H_2$. "Bridge" is a difficult problem for the mechanized transplanting of super rice. If the height of the transplanting trajectory is low, the transplanting arm will plant the tip of the previous seedling and the root of the subsequent seedling into the water field, creating a "bridge" phenomenon. Therefore, when the trajectory height is greater than the total height of the seedlings, the "bridge" phenomenon can be avoided. To prevent the occurrence of a "bridge" phenomenon during planting, set $H_2 \geq 300$ mm, serving as an optimization target to avoid seedlings forming a "bridge".

$$H_2 = y_{J_1(max)} - y_{J_1(min)} \tag{25}$$

The terms in the equation are defined as follows:

$H_2$ represents the height of the transplanting trajectory;

$y_{J_1(max)}$ signifies the ordinate of the highest point of the transplanting trajectory;

$y_{J_1(min)}$ designates the ordinate of the lowest point of the transplanting trajectory.

(8) The gearbox's elevation from the ground exceeds 25 mm. As shown in Figure 10, the distance from the ground to the gearbox is $l_2$, so we set $l_2 \geq 25$ mm as an optimization target to ensure the gearbox's elevation from the ground fulfills the requirements.

$$l_2 = 3l_{o_1o_2} + r2_{dm\,max} \tag{26}$$

$$y_{dm} = y_c - k_1 \tag{27}$$

The terms in the equation are defined as follows:

$r_{max}$ signifies the maximum polar coordinate radius of the planetary pitch curve;

$k_1$ signifies the seedling pot's height, which is typically considered to be 20 mm;

$y_{dm}$ signifies the vertical coordinate of the horizontal plane where the ground is situated;

$y_c$ signifies the ordinate of the lowest point of the relative motion trajectory;

$k_2$ signifies the sum of the planetary gear's addendum height and the minimum clearance between the planetary gear's addendum circle and the gearbox.

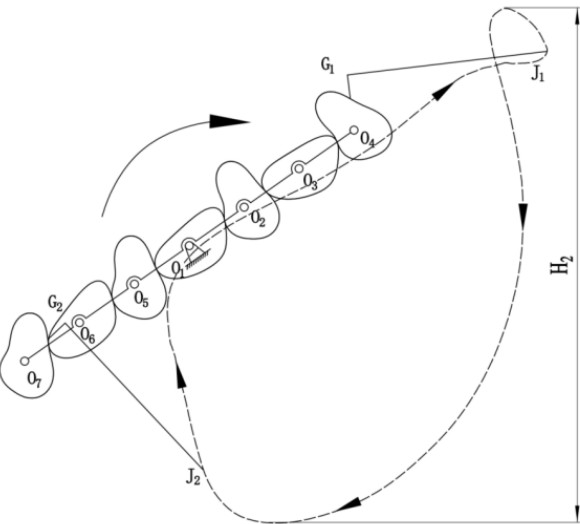

**Figure 9.** Schematic diagram of trajectory height.

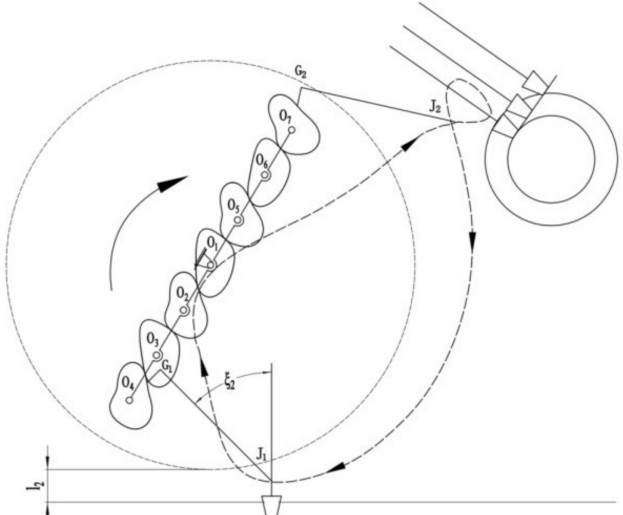

**Figure 10.** Schematic diagram of the distance between the gearbox and the ground.

(9) The transplanting arm should not push the seedlings. As illustrated in Figure 11, the intersection point of the straight line, where the planted rice seedling is located, and the trajectory is marked as point C. The vertical distance from the ground to point C is recorded as $l_{oc}$. The stem height (excluding leaves) of the seedlings at the transplanting stage approximates 80–100 mm, thus set $l_{oc} \geq 105$ mm as the optimization target for preventing the transplanting arm from pushing seedlings.

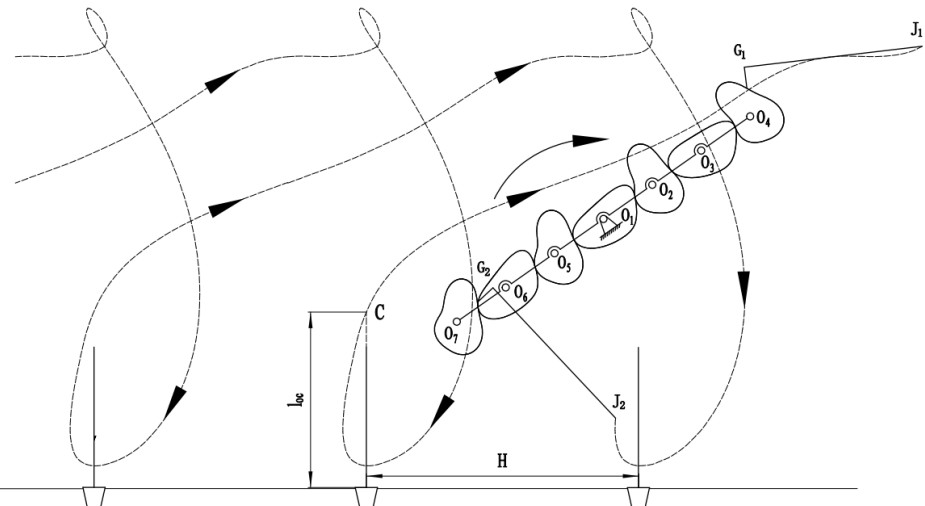

**Figure 11.** Schematic diagram of transplanting arm without pushing seedlings.

### 2.4.3. Parameter Determination

"Parameter-guided" heuristic optimization algorithm is an algorithm for solving multi-objective, multi-parameter, and strongly coupled optimization problems. After digitizing fuzzy objectives, automatic optimization can be achieved. The parameter optimization of the super rice transplanting mechanism is a complex optimization problem with multiple objectives, parameters, and nonlinearity. The "parameter-guided" heuristic optimization algorithm has solved the problem of parameter optimization for transplanting mechanisms. An optimization method that merges the "parameter-guided" heuristic optimization algorithm with manual fine-tuning has been employed [20]. This provides specific parameter values, as shown in Table 1. Here, $(r_i, \theta_i)$ denotes the control point of the non-circular gear pitch curve. $\delta_0$ is the initial installation angle of the transplanting arm. S represents the distance from the rotation center of the planetary wheel to the tip of the transplanting arm. H1 indicates the distance from the rotation center of the planetary wheel to the axis of the pusher rod. $\delta_0$ is the initial installation angle of the transplanting arm.

**Table 1.** Table of optimization results for parameter values.

| Input Parameters | Parameter Value/mm | Input Parameters | Parameter Value/° |
|---|---|---|---|
| r1 | 50 | θ1 | 31 |
| r2 | 8 | θ2 | 96 |
| r3 | 51 | θ3 | 34 |
| r4 | 67 | θ4 | 182 |
| r5 | −4 | θ5 | 285 |
| r6 | −43 | θ6 | 240 |
| r7 | 113 | θ7 | 175 |
| r8 | 50 | θ8 | 235 |
| r9 | 24 | θ9 | 280 |
| r10 | 10 | θ10 | 305 |
| r11 | 31 | θ11 | 287 |
| S | 150 | δ0 | −35 |
| H1 | 40 | φH0 | 35 |

### 2.5. Introduction of Relative Motion Trajectory

Figure 12 presents the relative trajectory of the specifically designed transplanting mechanism for super rice pot seedlings. The complete transplant trajectory adopts an "8" shape. Point A marks the initial position of the transplanting arm's sharp point. The segment from point A to point B indicates the seedling clamping preparation stage, during which the shifting fork remains static, the seedling clamping is naturally open, and the

transplanting arm progressively approaches the seedling stalk. The seedling clamping stage spans from point B to point C. In this phase, the shifting fork propels the seedling pusher to retract, squeezing the seedling clamping slices. As a result, the seedling clamping slices begin to close. When the clamping slices' sharp point reaches point C, they are entirely closed to grip the seedlings, completing the seedling clamping action. The stage of pulling out seedlings extends from point C to point D. In this phase, the shifting fork does not move, the pusher rod and the shell of the transplanting arm remain relatively static, and the seedling clamping stays closed, grasping the seedling and extracting it from the seedling pot. The interval from point D to point E designates the seedling delivery stage, during which the shifting fork remains immobile. The seedling pushing rod and the transplanting arm shell are relatively stationary, and the seedling clamping slices grip the seedlings and transport them to the planting position, simultaneously turning the seedlings to an upright position. Points E to F signify the planting stage. When the tip of the seedling clamping slices moves to point E, it reaches the planting position and begins to push the seedlings. The transplanting arm rotates relative to the cam, and the shifting fork prompts the pusher rod to extend out, prompting the seedling clamping slices to open quickly and the pusher to move downward to eject the pot seedlings. The components of the transplanting arm sustain this state until point F. To ensure that the seedlings are fully pushed out and descend into the field, when the sharp point of the seedling clamping slices reaches the F point position, the pushing action concludes. The reset stage encompasses points F to A, during which the pusher rod retracts upward to its initial position, driven by the shifting fork. Once the sharp point of the seedling clamping slices reaches point A, the transplanting arm returns to its starting position, completing the reset and preparing for the next seedling pickup.

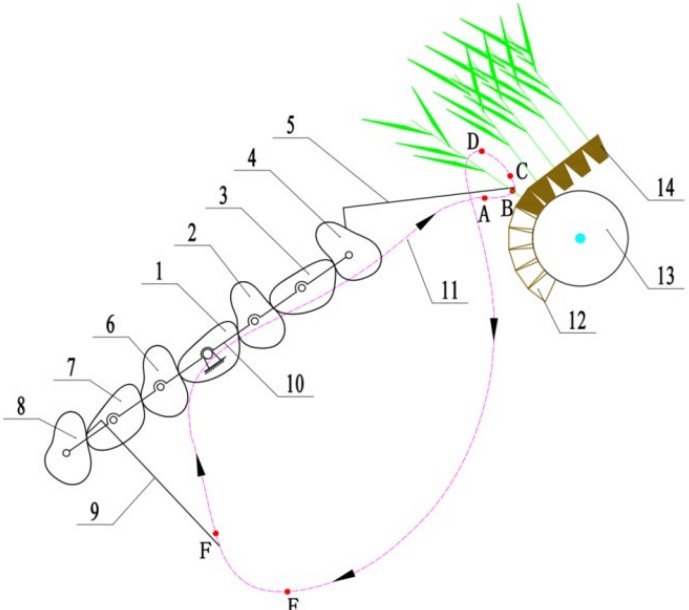

**Figure 12.** Relative motion trajectory diagram. 1. Sun gear; 2. Intermediate gear I; 3. Intermediate gear II; 4. Planetary gear I; 5. Transplanting arm I; 6. Intermediate gear III; 7. Intermediate gear IV; 8. Planetary gear II; 9. Transplanting arm II; 10. Planetary carrier; 11. relative motion trajectory; 12. seedling tray; 13. conveying roller; 14. pot seedling.

## 3. Results and Discussion

*3.1. Experimental Research*

Virtual Test

A virtual prototype of the super rice PSTM was constructed based on the optimized mechanism parameters, and a virtual simulation environment was established in ADAMS, as shown in Figure 13.

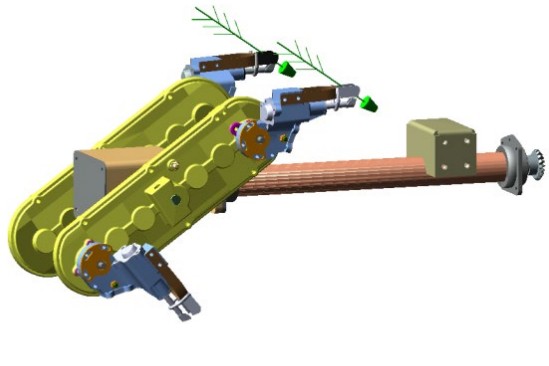

**Figure 13.** Virtual prototype.

The simulation results for the relative motion trajectory of the transplanting mechanism are displayed in line 1 of Figure 14a. Figure 14b provides a comparison between the virtual simulation trajectory and the theoretical trajectory. It is evident from the comparison that the relative motion trajectory, obtained via virtual prototype simulation, and the motion trajectory shaped by the theoretical model align closely. The maximum deviation distance is a mere 0.3 mm, and the trajectories are essentially identical, thereby validating the accuracy of both the kinematics theoretical model and the virtual prototype structure.

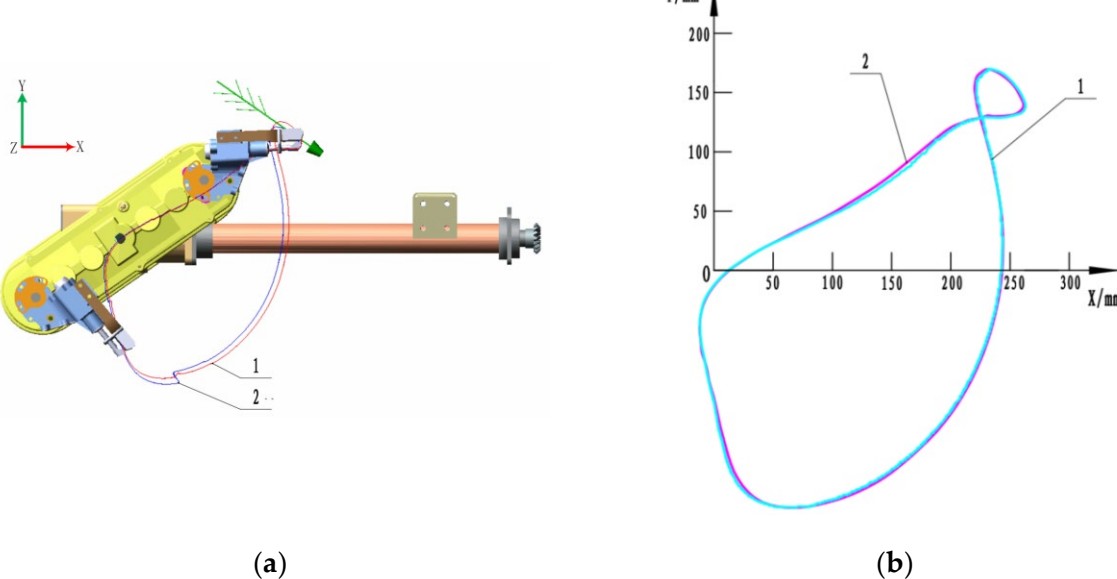

(**a**)                                                                                                    (**b**)

**Figure 14.** Comparison of relative motion trajectory results. (**a**) The relative motion trajectory of the virtual prototype. 1. Trajectory of the seedling clamping slices' sharp point. 2. Trajectory of the pusher's sharp point. (**b**) Comparison chart of simulated trajectory and theoretical trajectory. 1. Virtual simulation trajectory; 2. theoretical model trajectory.

To ascertain if the seedling pushing device can expel the seedlings at the planting point, an analysis of the trajectory of the seedling pushing device was also conducted, as shown in line 2 of Figure 14. Given that seedling pushing is a process, the seedlings must be pushed before the trajectory's lowest point. When the tip of the transplanting arm is removed from the trajectory's lowest point, the planetary carrier's corner initiates the seedling push 7–10° earlier, improving the seedlings' uprightness. According to Figure 15, the time difference between the seedling clamping slices' tip's lowest point and the pusher's starting point is 0.022 s. Coupled with the planetary carrier's rotation speed of 1 r/s (simulation data), it

is determined that the transplanting arm begins to push the seedlings 7.92° ahead of the lowest point, which fulfills the design requirements.

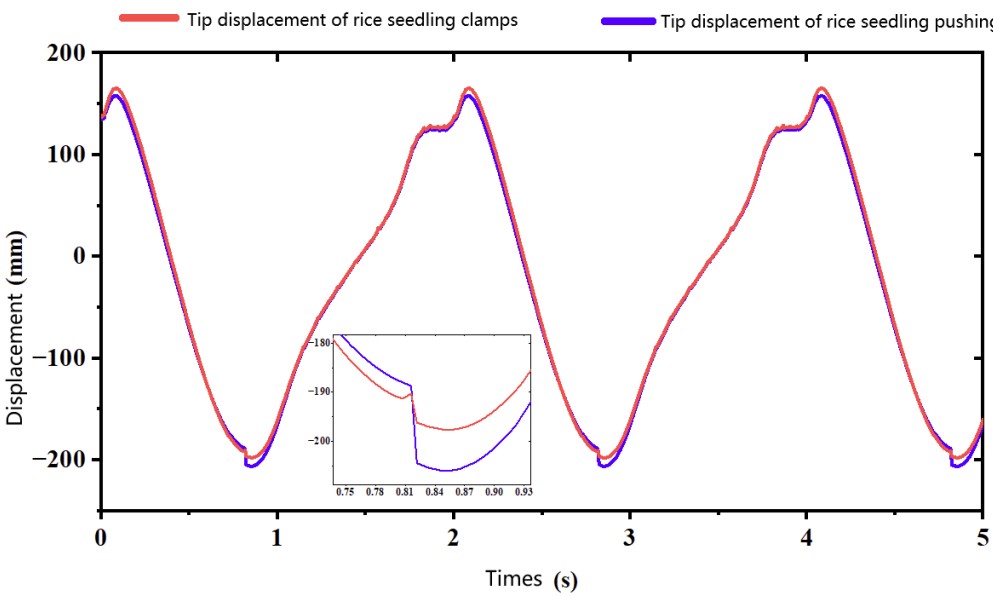

**Figure 15.** Y-direction displacement diagram of seedling clip and pusher.

An absolute motion trajectory simulation of the virtual prototype was conducted, and the result is shown in Figure 16a. It aligns closely with the transplanting trajectory formed by the theoretical model in the optimization auxiliary software, thereby validating the accuracy of the kinematics theoretical model and the structure of the virtual prototype.

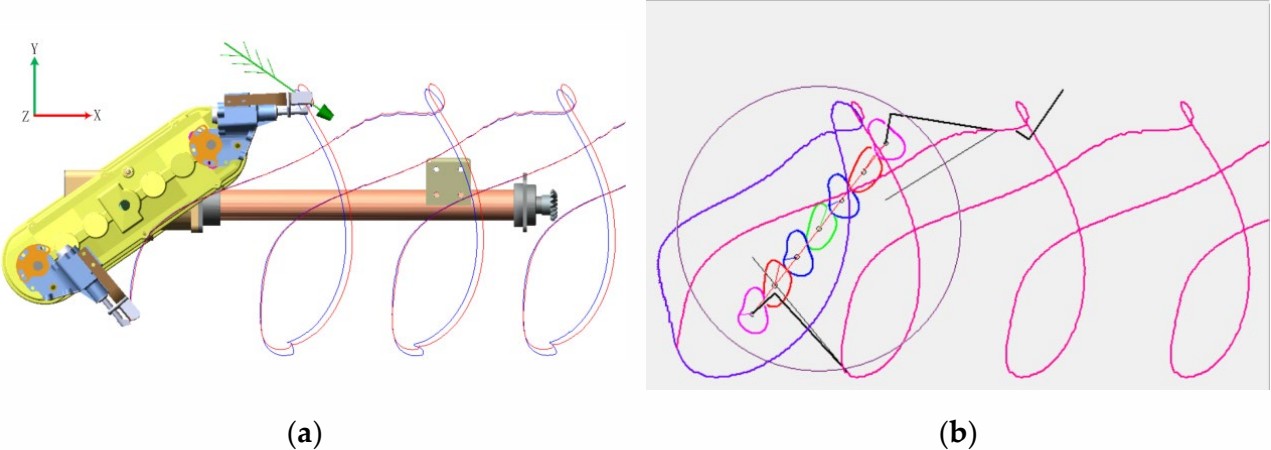

(**a**)               (**b**)

**Figure 16.** Comparison of absolute motion trajectory results. (**a**) The absolute trajectory of the virtual prototype. (**b**) Absolute trajectory of the theoretical model.

### 3.2. Trajectory and Attitude Verification Test

### 3.2.1. Development of Test Bench

To analyze the practical operating performance of the devised transplanting mechanism, a test bench was developed for the transplanting mechanism of rice pot seedlings. The constructed transplanting mechanism test bench is depicted in Figure 17. The mechanical portion of the transplanting mechanism test bench primarily comprises the frame, the transplanting mechanism's transmission component, the seedling box, the horizontal seedling delivery mechanism, and the longitudinal seedling delivery mechanism. The frame, positioned at the bottom of the test bench, serves to install components such as the seedling box and the transplanting mechanism's transmission box. The transplanting

mechanism's transmission device is located at the front of the frame and is utilized to install and drive the designed transplanting mechanism. The seedling box, situated in the middle of the frame, is utilized to house the seedling tray and work in unison with the transplanting mechanism to complete the seedling picking process. The horizontal seedling feeding mechanism is installed at the seedling box's base, which drives the seedling box to shift left and right, enabling horizontal seedling feeding. The longitudinal seedling feeding mechanism, fitted on the seedling box's side, propels the seedling tray on the seedling box to move downward, thereby realizing longitudinal seedling feeding.

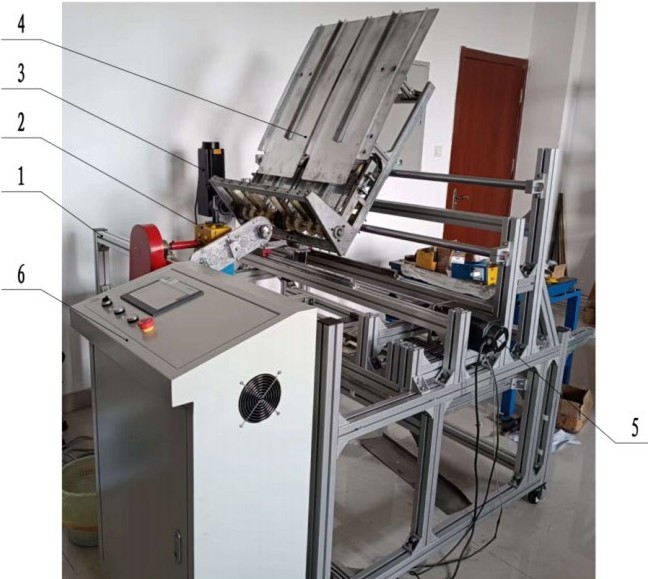

**Figure 17.** The physical picture of the transplanting mechanism test bench. 1. Frame; 2. transmission device of transplanting mechanism; 3. longitudinal seedling delivery mechanism; 4. seedling box; 5. horizontal seedling delivery mechanism; 6. control cabinet.

### 3.2.2. High-Speed Camera Test

A high-speed camera was used to capture the rotation process of the transplanting mechanism, and the relative motion trajectory of the seedling clamping slice's sharp point over one cycle was extracted. Figure 18a displays the relative motion trajectory of the transplanting arm's sharp point as it clamps the seedlings, captured by the high-speed camera. Figure 18b presents a comparison between the trajectory captured by the high-speed camera and the theoretical motion trajectory. Upon comparing the two, it is evident that a slight deviation exists between the relative motion trajectory of the physical prototype and the trajectory derived from the theoretical analysis during the seedling picking process. This discrepancy is attributed to an increase in the angular acceleration of the transplanting arm's rotation at the seedling picking position and not to the backlash of the circular gear transmission, which would cause vibration. Although a slight deviation between the prototype's trajectory and the theoretical trajectory is observed, the maximum deviation distance is merely 2.3 mm. The two trajectories are largely congruent, thereby satisfying the design requirements.

A high-speed camera was used to capture the posture of the transplanting arm at the crucial positions of seedling picking and pushing, as illustrated in Figure 19. An analysis was conducted, comparing the seedling picking angle, seedling pushing angle, and angle difference of the transplanting arm as captured by the high-speed camera with the corresponding theoretical values in the optimization software. The objective was to determine whether the posture of the transplanting arm at these critical positions of picking and pushing seedlings fulfills the design requirements. Table 2 shows that the actual attitude parameters of the transplanting arm slightly deviate from the theoretical parameters in the optimization design software, but they remain within the acceptable

design parameters. The differences observed can be attributed to several factors: (1) The gearbox of the transplanting mechanism employs a three-stage gear transmission, and there is a backlash between the teeth. This affects the rotation angle of the transplanting arm, which is installed at the end of the gearbox, relative to the planetary frame, resulting in a variation between the seedling picking angle and the seedling pushing angle. (2) There is a discrepancy between the initial installation angle of the transplanting arm and the theoretical initial installation angle, contributing to the observed differences. (3) The process of seedling picking and pushing is dynamic. The positions of picking and pushing seedlings as captured by the high-speed camera deviate from the theoretical picking and pushing positions of seedlings in the optimization design software.

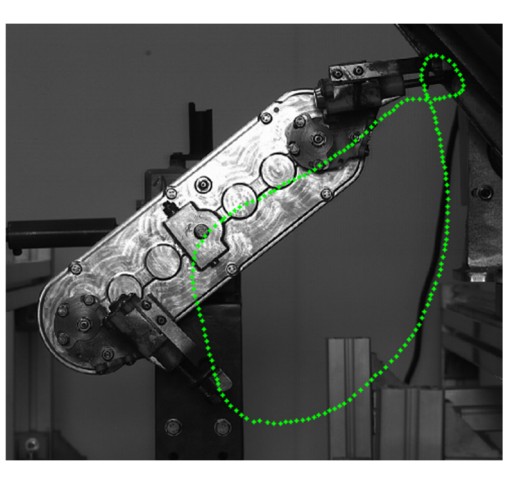

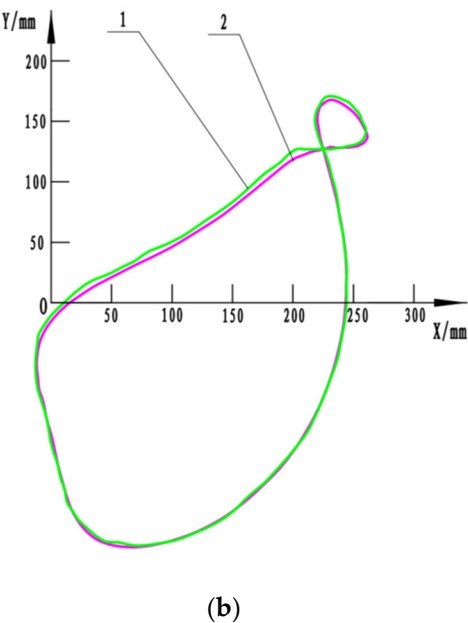

(**a**)　　　　　　　　　　　　　　　　　　(**b**)

**Figure 18.** Comparison of relative motion trajectories. (**a**) Relative motion trajectory of high-speed camera. (**b**) Comparison of high-speed camera trajectory and theoretical trajectory. 1. High-speed camera trajectory; 2. theoretical model trajectory.

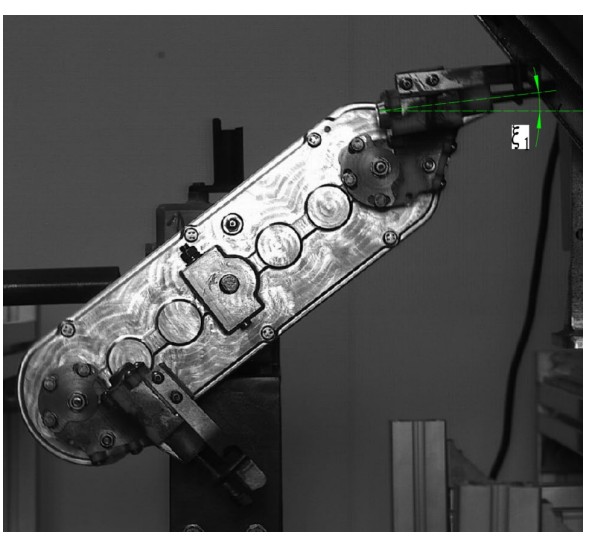

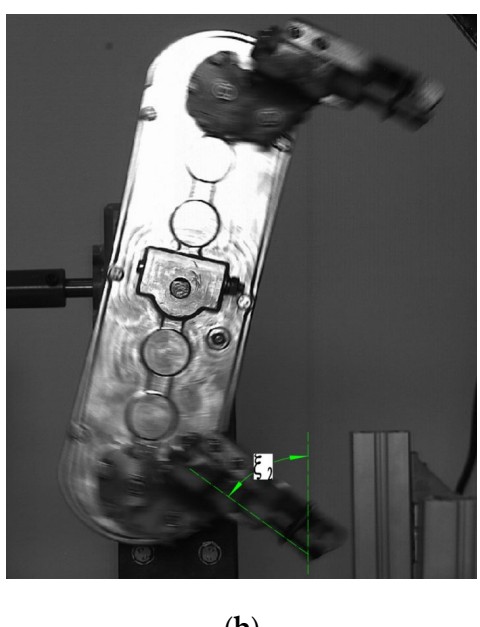

(**a**)　　　　　　　　　　　　　　　　　　(**b**)

**Figure 19.** Posture analysis of the transplanting arm. (**a**) Seedling picking position. (**b**) Seedling pushing position.

**Table 2.** Posture analysis table of key points of the transplanting arm.

| | Seedling Picking Angle ($\xi_1$) | Seedling Pushing Angle ($\xi_2$) | Angle Difference ($\xi_2-\xi_1$) |
|---|---|---|---|
| Design Requirements | $-5$–$10°$ | $45$–$70°$ | $50$–$60°$ |
| Theoretical Design | $8°$ | $62°$ | $54°$ |
| Physical Prototype | $6.62°$ | $59.32°$ | $52.7°$ |

### 3.3. Seedling Test

The cultivated rice pot seedlings are shown in Figure 20. The seedling raising period is 35 days, and the specification of the seedling tray and seedling hole is $14 \times 29$. There are 14 seedling pots horizontally and 29 seedling pots vertically. The total number of single tray seedling holes is 406, the depth of the seedling holes is 19 mm, the length of the tray is 582 mm, and the width is 282 mm.

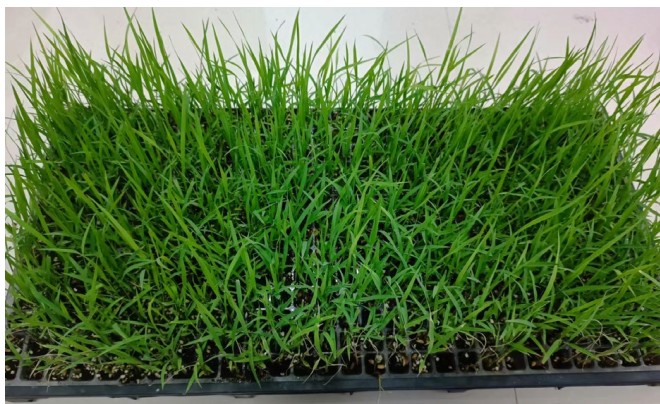

**Figure 20.** Experimental pot seedlings.

To analyze the seedling picking performance of the transplanting mechanism, different rotational speeds were set for its planetary frame, as demonstrated in Figure 21. Two trays of cultivated seedlings were selected, accounting for a total of 812 seedling holes. These two sets of seedlings were placed on the test bench's seedling box, and the transplanting mechanism was adjusted to its initial position, with the planetary frame set to rotate at 90 r/min. After all seedlings from the two trays were picked up, the test bench operation was halted, and the number of successfully picked seedlings was counted to calculate the seedling success rate. The same procedure was repeated twice more but this time with the planetary frame rotating at 100 r/min and 110 r/min, respectively. The outcomes of these three experiments were then compared, as shown in Table 3.

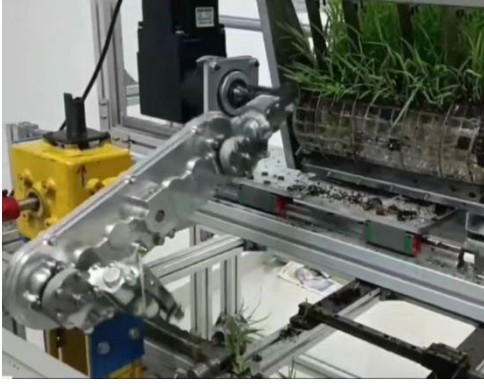 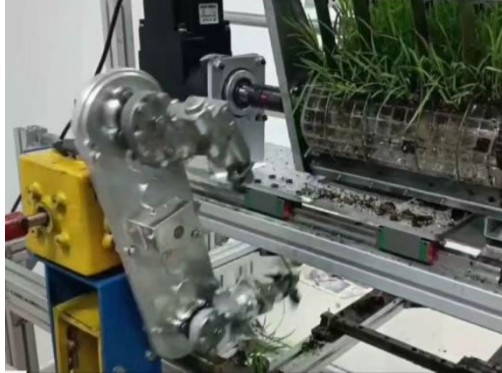

**Figure 21.** Seedling picking test.

**Table 3.** Analysis of Seedling Picking Performance.

| Rotating Speed/rpm | Number of Seedlings in Seedling Tray/Plant | The Number of Seedlings Taken out/Plant | Seedling Success Rate/% | Injury Rate/% |
|---|---|---|---|---|
| 90 | 812 | 780 | 96 | 1.5% |
| 100 | 812 | 788 | 97 | 1.8% |
| 110 | 812 | 756 | 93 | 2.2% |

The extent of damage inflicted on the seedlings by the transplanting mechanism during the transplanting process serves as another critical performance evaluation metric. As the designed transplanting mechanism employs a stem clamping method for picking seedlings, the damage mainly stems from the stem pinching action of the clamping pieces during the picking phase. The employed test method involved observing the damage to the contact area between the seedling clamping slices and the seedling stem under an industrial microscope with a magnification range of 21–135 times, as depicted in Figure 22.

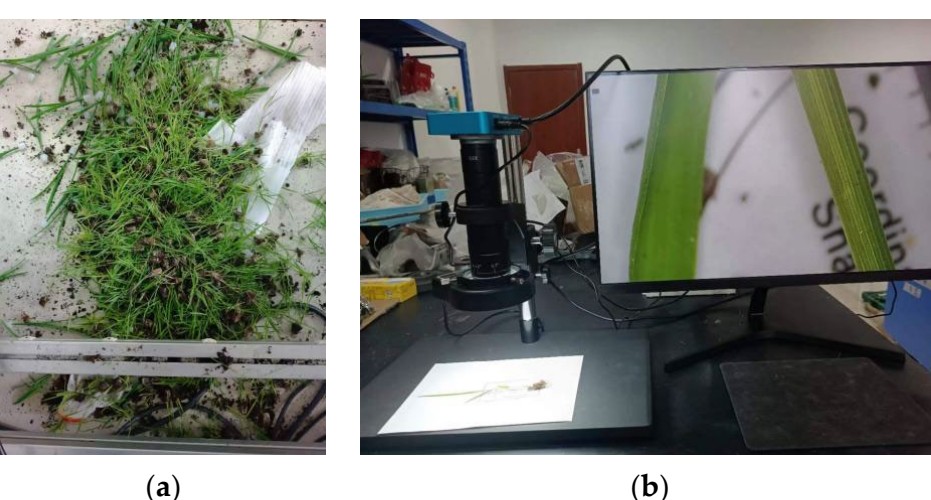

(**a**)        (**b**)

**Figure 22.** Damage detection of rice seedling stalks. (**a**) Seedlings were picked up from the bench test. (**b**) Rice seedling damage detection device.

Based on the degree of damage to the stems measured, the rice seedlings' stem damage was categorized into three levels: slight, moderate, and severe damage. Slight damage involves minor surface indentations on the seedlings, while moderate damage indicates the presence of cracks on the seedling stalk's surface. Severe damage refers to a break in the seedling stalk. These are illustrated in Figure 23. Given the team's years of experience in rice transplanting, slight damage has a negligible impact on subsequent seedling growth, while moderate and severe damage significantly affect seedling development and growth. Thus, the count of moderately and severely damaged seedlings was recorded as the number of injured seedlings. Subsequently, the damage rate of seedlings picked up at different rotational speeds was assessed and analyzed, as presented in Table 3. The causes of seedling damage were analyzed, mainly including seedling growth not in the center of the hole plate, some root systems growing and knotting from the bottom of the hole plate, and trajectory and posture deviation caused by manufacturing and installation errors of the mechanism.

In accordance with the technical specifications for the quality evaluation of rice pot seedling planting machines, clamping rice planting machinery must adhere to a missed planting rate of 3% and a seedling injury rate of less than 2% to meet the technical specifications. Upon comparison, it was observed that when the transplanting mechanism operates at a rotational speed of 100 r/min, with a transplanting efficiency of 200 plants/min, the success rate of seedling picking is high, and the seedling damage rate is low. These conditions align with industry standards and optimally meet practical transplanting needs.

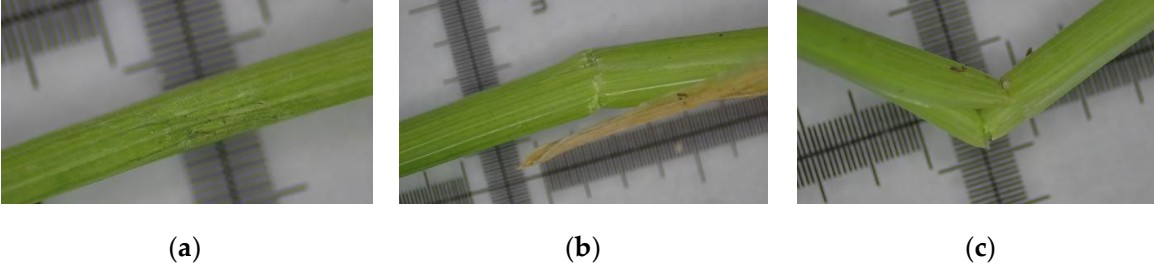

<div align="center">(<b>a</b>)     (<b>b</b>)     (<b>c</b>)</div>

**Figure 23.** Grading diagram of seedling damage degree. (**a**) Slight damage. (**b**) Moderate damage. (**c**) Severe damage.

### 4. Conclusions

1. A differential-speed rotary super rice pot seedling transplanting mechanism (PSTM) was proposed, and through the utilization of bespoke super rice PSTM optimization design software, a set of mechanism parameters was optimized to align the transplanting arm with the super rice PSTM. This trajectory and posture required for pot seedling transplantation is significantly beneficial for the mechanized transplanting of super rice pot seedlings.

2. Based on the kinematics model of the differential-speed rotary super rice PSTM, the optimization design software was developed, and a set of mechanism parameters that satisfied the super rice pot seedling transplanting requirements was optimized. Using high-speed camera technology, the trajectory and posture of the transplanting mechanism were verified. This validation illustrates the correctness of the theoretical analysis and design method of the transplanting mechanism.

3. A rice PSTM test bench was developed, and the seedling picking test was successfully conducted. The test indicated that when the transplanting efficiency is 200 plants/min, the seedling extraction success rate is 97%, and the seedling injury rate is 1.8%. This seedling picking performance complies with the quality and technical specifications of rice pot seedling planting machines and has practical application value.

**Author Contributions:** Conceptualization, M.Z., Z.W. (Zhaoxiang Wei), Z.W. (Zeliang Wang), H.S., G.W. and J.Y.; methodology, M.Z., Z.W. (Zhaoxiang Wei), Z.W. (Zeliang Wang), H.S., G.W. and J.Y.; software, Z.W. (Zhaoxiang Wei) and Z.W. (Zeliang Wang); validation, M.Z., Z.W. (Zhaoxiang Wei), Z.W. (Zeliang Wang), H.S., G.W. and J.Y.; formal analysis, M.Z., Z.W. (Zhaoxiang Wei), Z.W. (Zeliang Wang), H.S., G.W. and J.Y.; investigation, Z.W. (Zhaoxiang Wei) and Z.W. (Zeliang Wang); resources, M.Z., Z.W. (Zhaoxiang Wei), Z.W. (Zeliang Wang), H.S., G.W. and J.Y.; data curation, M.Z., Z.W. (Zhaoxiang Wei), Z.W. (Zeliang Wang), H.S., G.W. and J.Y.; writing—original draft preparation, Z.W. (Zhaoxiang Wei) and Z.W. (Zeliang Wang); writing—review and editing, M.Z. and J.Y.; visualization, M.Z., Z.W. (Zhaoxiang Wei), Z.W. (Zeliang Wang), H.S., G.W. and J.Y.; supervision, M.Z. and J.Y.; project administration, M.Z. and J.Y.; funding acquisition, M.Z. and J.Y. All authors have read and agreed to the published version of the manuscript.

**Funding:** This study was financially supported by the National Natural Science Foundation of China (Grant No. 52005221), Jiangsu Agriculture Science and Technology Innovation Fund (Grant No. CX(22)3089), China Postdoctoral Science Foundation (Grant No. 2021M691315), Key R&D Plan of Zhenjiang City—Modern Agriculture (Grant No. NY2023003), Natural Science Foundation of Jiangsu Province (Grant No. BK20200897), Key Laboratory of Modern Agricultural Equipment and Technology (Jiangsu University), High-Tech Key Laboratory of Agricultural Equipment and Intelligence of Jiangsu Province, and Priority Academic Program Development of Jiangsu Higher Education Institutions (Grant No. PAPD-2018-87).

**Institutional Review Board Statement:** Our studies did not involve humans or animals.

**Informed Consent Statement:** Not applicable.

**Data Availability Statement:** The data presented in this study are available on request from the corresponding author. The data are not publicly available due to our laboratory privacy data protection.

**Conflicts of Interest:** The authors declare no conflict of interest.

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
