# Peer review of "Design and Experimental Investigation of a Transplanting Mechanism for Super Rice Pot Seedlings"

_agriculture, doi:10.3390/agriculture13101920_

Round 1

Reviewer 1 Report

The author introduces a differential-speed rotary mechanism for transplanting super rice pot seedlings. The developed mechanism can enable the transplanting arm to reproduce the special trajectory and posture required for transplanting super rice pot seedlings. The kinematic model was established and an optimization design software was developed to obtain the parameters required for transplanting super rice pot seedlings.Finally, a seedling pickup experiment was conducted, and when the seedling pickup efficiency was 200 times/min, the success rate of seedling pickup was 97%, and the seedling injury rate was 1.8%, which met the requirements for rice pot seedling transplantation. This study provides a reference for mechanized transplanting of super rice pot seedlings, and has novelty, practicality, and theoretical value.

There are some problems, which must be solved before it is considered for publication. If the following problems are well-addressed, this reviewer believes that the essential contribution of this paper are important for agricultural engineering.

1. In section 2.3, “where the non-circular gear adopts a Bezier gear and its gear pitch curve forms a Bezier curve”, Why did the author use the Bezier curve? What are its advantages?

2. In section 2.4.3, “Employed an optimization method that merges the ‘parameter-guided’ heuristic optimization algorithm with manual fine-tuning”, The author needs to further explain why this algorithm should be used.

3. The transplanting mechanism proposed by the author is different from existing institutions, and what are the specific advantages and connections with super rice transplanting? It should be clearly stated in the text.

4. In section 2.4.2, The character label in Figure 7 is too small, which affects reading.

5. In section3.1.1, The words in the upper left corner of Figure 13 are unclear, does it have any meaning?

6. The meaning of the first sentence and the next sentence in the abstract is not coherent and should be revised and corrected.

No comments on the quality of English language.

Reviewer 2 Report

Dear Authors,

First of all I want to congratulate for your hard work. Below you can find my comments.

1.       In my opinion, some letter should be in italic type. Such as, “r1” in line 163 (equation 2) should be “r1”; “r2” in line 163 (equation 2) should be “r2”; and also in the figures. Other similar issues should be checked by the author themselves.

2.       The writing of variables is inconsistent. In line 158 the variable is written as “r1(i)”, while in 163 the variable is written as “r1(i)”. Other similar issues should be checked by the author themselves.

3.       I do not recommend using MathType to edit formulas, unless the author can adjust the font size of the equations appropriately. This issue leads to the following problems:

(1)   The spacing between line 168 and 169 is not equal to other parts due to formula font size issues. There are the same issues in the line space of line 164 and 165, line 164 and 165, etc.

(2)   The font size of the equations are not the same. Such as equation 5 and equation 6.

(3)   Some letter is not displayed completely. Such as “η” in line 164.

4.       The experiment is well structured and described. However, I consider it necessary to present field tests or soil bin test, not just bench tests. Because quality indicators such as planting depth and row spacing for rice planting are difficult to measure in the laboratory, they are important parameters for evaluating the PSTM.

5.       The “material and methods” is not adequately described. Important parameters such as rice variety, seedling age and soil conditions for potted seedlings have not been mentioned in “Seedling test” part, as they can affect the results of missed planting rate, seedling injury rate and Seedling success rate.

Reviewer 3 Report

The author introduces a differential-speed rotary mechanism for transplanting super rice pot seedlings. This transplanting mechanism can provide the required trajectory and posture for transplanting super rice. Subsequently, a kinematic model was established, virtual simulation was conducted, and optimization design software was developed to solve the optimal parameters. Finally, a seedling pickup experiment was conducted, and based on the experimental results, it was found that when the seedling pickup efficiency was 200 times/min, the success rate of seedling pickup was 97%, and the seedling injury rate was 1.8%, meeting the industry requirements for rice transplantation. This study provides a reference for the mechanized transplanting of super rice, and has innovation, practicality, and theoretical value.

However, they need to address the following concerns before the paper gets published.

1.In 2.4.1, the introduction to optimizing software functions is not detailed enough, and the author should further explain.

2.In 2.4.2 (3), the author should further elaborate on why the selection of seedling angle should be between -5 ° and -10 °.

3.In Figure 7 of 2.4.2, some annotations are unclear and should be adjusted and revised.

4.In 2.4.3 (7). What does "bridging" mean? What is the impact on transplanting super rice? The author should provide a detailed explanation.

5.In 3.2.2, the angle mark in Figure 19 (b) is too small, which affects reading.

Reviewer 4 Report

The author be aimed at the issue that current rice transplanters cannot meet the special trajectory and posture required for super rice transplantation, a differential-speed rotary mechanism for transplanting super rice pot seedlings was proposed. The author established a kinematic model, conducted theoretical calculations, and developed an optimization design software. Using this software, a set of parameters that met the requirements for transplanting super rice were obtained. Finally, the author conducted a seedling pickup experiment and found that the best effect was achieved when the seedling pickup efficiency was 200 times/min. The seedling pickup success rate of 97% and a seedling injury rate of 1.8%, which met the industry requirements for rice transplantation. This study provides a reference for the transplantation of super rice, which is innovative and worthy of promotion and application.

The author should make modifications to the following questions.

1. The author has developed a rice pot seedling transplanting machine for super rice transplanting. Compared with conventional rice transplanting, what are the characteristics of super rice transplanting and what are the new design requirements for transplanting mechanism design?

2. In 2.4.1, what are the main parameters that can be inputted and adjusted in the optimization software developed by the author? How are these parameters determined? Further clarification should be provided.

3. During the transplant experiment, the experimental conditions were not clearly stated. How long are seedlings cultivated? What are the characteristics? The author should explain.

4. When analyzing the situation of damaged seedlings, what are the main causes of damaged seedlings? The author should analyze it.

5. In section 3.3, “The extent of damage inflicted on the seedlings by the transplanting mechanism during the transplanting process serves as another critical performance evaluation metric. ”, The author should briefly explain the standards mentioned in the article that meet the requirements of the rice transplanting industry.

6. The annotation of some illustrations in the article is not clear, which affects reading and should be revised and corrected.

Round 2

Reviewer 2 Report

Dear Authors,

Once again, I would like to congratulate you on the work you have done. In my opinion, the improvements you have made to the manuscript are fully satisfactory.

I suggest that in future research tests, the authors can refer to the industry standard NY/T 3013-2016.

Kind Regards.